

# Improving the Accuracy of Model-based Quantitative NMR

Yevgen Matviychuk[1], Ellen Steimers[2], Erik von Harbou[2,3], and Daniel J. Holland[1]

[1]University of Canterbury, Private Bag 4800, Cristchurch 8140, New Zealand
[2]Technische Universität Kaiserslautern, Erwin-Schrödinger-Straße 44, Kaiserslautern 67663, Germany
[3]Current address: BASF SE, Research and Development, Ludwigshafen, Germany

**Correspondence:** Yevgen Matviychuk (eugene.matviychuk@canterbury.ac.nz)

**Abstract.** Low spectral resolution and extensive peak overlap are the common challenges that preclude quantitative analysis of NMR data with the established peak integration method. While numerous model-based approaches overcome these obstacles and enable quantification, they intrinsically rely on rigid assumptions about functional forms for peaks, which are often insufficient to account for all unforeseen imperfections in experimental data. Indeed, even in spectra with well separated peaks whose integration is possible, model-based methods often achieve suboptimal results, which in turn raises the question of their validity for more challenging datasets. We address this problem with a simple model adjustment procedure, which draws its inspiration directly from the peak integration approach that is almost invariant to lineshape deviations. Specifically, we aim to recover all useful signal left in the residual after model fitting and use it to adjust the intensity estimates of modelled peaks. We propose an alternative objective function, which we found particularly useful for correcting imperfect phasing of the data – a critical step in the processing pipeline. Application of our method to the analysis of experimental data shows the accuracy improvement of 20-40% compared to the simple least squares model fitting.

## 1 Introduction

Proposed thirty years ago (Miller and Greene, 1989; Bretthorst, 1990; Chylla and Markley, 1995), model-based approaches for quantitative NMR data analysis (qNMR) are getting wider acceptance as an effective alternative to the established peak integration (Kriesten et al., 2008; Krishnamurthy et al., 2017; Kern et al., 2018). Based on the idea that an experimental spectrum can be represented as a collection of parametric lineshapes, e.g. Lorentzians with certain positions, widths, and heights, they offer a principled mechanism to resolve overlapping peaks and are less susceptible to noise (Matviychuk et al., 2017). By adjusting its parameters, the model is fitted to match experimental data, which eventually determines the sought concentrations of chemical species in the analysed sample. The reduction of a spectrum to a frequency-intensity table of peaks (Krishnamurthy, 2013) allows for easier automation of post-processing tasks and simplifies the analysis of large arrayed datasets (Kriesten et al., 2008; Alsmeyer et al., 2004). Finally, quantum mechanical formulations minimize the number of free model parameters and are inherently invariant with respect to the spectrometer field strength (Kuprov et al., 2007; Tiainen et al., 2014; Dashti et al., 2017); they enable the analysis of highly complex low-resolution spectra acquired on medium-field benchtop instruments and are found successful in modern practical applications (Matviychuk et al., 2019).





There have been proposed numerous model-based approaches to the problem of qNMR formulated either in the time (Vanhamme et al., 2001; Krishnamurthy, 2013; Rubtsov and Griffin, 2007) or the frequency domain (Mierisová and Ala-Korpela, 2001; Poullet et al., 2008). Notably, the latter typically depend upon phase and baseline correction of the spectra before fitting signal models to them (Cobas et al., 2008; Kriesten et al., 2008). In contrast, time-domain methods that work with the FID signal are often regarded as being able to lift this requirement (Krishnamurthy, 2013). However, we note that when the model fitting is performed in the least squares sense – as done most commonly – both variants of the problem formulations are equivalent and result in the same solution. Thus, even though explicit data preprocessing steps can be often obviated by time-domain methods, they inevitably include the phasing parameters in a certain form, either as angles of complex-valued intensity estimates for separate resonances (Krishnamurthy, 2013; Rubtsov and Griffin, 2007; Kung et al., 1983) or as independently optimized parameters of a linear phasing model (Matviychuk et al., 2017). On the other hand, the phasing parameters can also be estimated from the complex-valued frequency domain data (Sokolenko et al., 2019). Similarly, the baseline effects that are often observed over wide spectral ranges, appear in the leading time points of the original FID signal. Hence, these distortions also need to be taken into account in the time domain analysis, either by masking or weighting the early time samples.

Despite numerous advantages, model-based qNMR is often found suboptimal in seemingly easy cases: when peaks in the spectrum are well resolved, and the signal-to-noise ratio (SNR) is suffciently high, the peak integration after careful phase and baseline correction typically achieves higher quantification accuracy, as we observe later in Sec. 4.1. This can be explained by the high sensitivity of most model-based qNMR algorithms to any unforeseen distortions in the experimental data, such as imperfections of peak shapes and their deviations from the assumed ideal Lorentzians. Indeed, model misspecification and its inability to faithfully represent the data biases the estimates of concentrations along with the associated uncertainties (White, 1981, 1982; Grünwald and van Ommen, 2017); this produces misleading results becoming one of the major points of criticism of model-based qNMR. To overcome this obstacle, several generalizations of the peak lineshape function have been proposed over time, most notably the Voigt lineshape (Humlíček, 1982; Marshall et al., 1997; Bruce et al., 2000) and other combinations of Lorentzian and Gaussian terms (Kriesten et al., 2008; Schoenberger et al., 2016). Nevertheless, peak shape deviations in experimental spectra can often be very hard to model explicitly within the parametric framework, as they typically reflect multiple independent physical processes, such as diffusion, magnetic field inhomogeneity, higher-order coupling effects, etc. Reference deconvolution methods (Morris et al., 1997; Metz et al., 2000; Osorio-Garcia et al., 2011) offer an effective mechanism to eliminate complex distortion patterns common for all peaks in the spectrum, e.g. arising due to the lack of shimming. However, they can not easily address possible differences in shapes of separate peaks, for example as a result of small long-range couplings, whose effects become even more noticeable at lower magnetic field strengths.

Since model-based qNMR is the only viable option for the analysis of complicated spectra with multiple overlapping peaks, it is of utmost importance to develop accurate algorithms that are robust to possible model misspecifications. The main goal of this work is to bridge the performance gap between the peak integration and model-based qNMR by combining the strength of both approaches. Specifically, after fitting a model to the data, we observe that the residual – instead of being purely noise – often contains non-stochastic elements pertinent to the useful NMR signals. We propose to explicitly incorporate this unaccounted remainder into the model-based analysis, as would have been done with peak integration. On the other hand,





neither of the existing phase and baseline correction methods take into account prior information about the studied system, which is conveniently employed in our approach in the form of an adjustable model. As a result, our alternative optimization procedure achieves better baseline and phase correction than the usual least-squares model fitting and improves model-based quantification of both well-resolved and overlapped data.

In the next section, we briefly review the main idea of model-based qNMR and introduce the notation for the problem of estimating the concentrations of components in a mixture. We then proceed by studying the weaknesses of the traditional least-squares model fitting and propose our alternative optimization criterion. Section 3 describes the setup for our simulations and laboratory experiments; their results are presented in Section 4.

## 2 Theory

This section provides a theoretical background for our method. First we review the general principle of qNMR: given a mixture of known chemical species, we are set to estimate their unknown relative concentrations (mole fractions) using the NMR data. In model-based qNMR, an ideal model that represents the studied mixture is fitted to the experimental data, and its found optimal parameters – specifically the intensities of model components – determine the estimates of concentrations of chemical species. Here we discuss the consequences of model misspecification, and propose our model adjustment method to improve the accuracy of quantification.

### 2.1 Overview and the main idea of model-based qNMR

We choose to formulate our method in the frequency domain using real-valued spectra. Even though discarding the imaginary counterpart of complex-valued data entails reduction of SNR by a factor of 2, this will allow us to develop an adjustment algorithm for our model-fitting method inspired by the peak integration, which traditionally operates with real-valued spectra. Thus, an experimental spectrum is obtained using the discrete Fourier transform of the acquired FID followed by the usual first-order phase correction with parameters $\varphi_0$ and $\varphi_1$. It is formally represented as an $n \times 1$ column vector $\mathbf{y} = Re\left[\mathcal{F}\left(\mathbf{y}_{\mathrm{T}}\right)e^{-i(\varphi_0 + \varphi_1 \mathbf{f})}\right]$, where $\mathbf{f} = \left[-\frac{1}{2\Delta t} \leq f \leq \frac{1}{2\Delta t}\right]$ is the vector of frequency values corresponding to the particular sampling (dwell) time of the FID, $\Delta t$.

Next we define a model matrix $\mathbf{Z}$ whose columns contain signature spectra for all $K$ analyzed chemical species evaluated on the same frequency grid $\mathbf{f}$. A typical signature model is a combination of $P$ elemental peaks with different chemical shifts, widths, and intensities $b_p$:

$$z_k\left(f\right) = \sum_{p=1}^{P} b_p u_p\left(f|f_p, \alpha_p\right). \tag{1}$$

Here $u_p\left(f|f_p, \alpha_p\right)$ defines an ideal Lorentzian peak with central frequency $f_p$ and full width at half maximum $\frac{\alpha_p}{\pi}$ (both expressed in Hz) evaluated at the frequency $f$,

$$u_p\left(f|f_p, \alpha_p\right) = \frac{\alpha_p \Delta t}{\left[2\pi\left(f - f_p\right)\right]^2 + \alpha_p^2}. \tag{2}$$





We note that $f_p = B_0 \delta_p - f_0$, where $B_0$ and $f_0$ are the operating frequency of spectrometer in MHz and the spectral offset respectively, which are used to convert the frequency units of the chemical shift $\delta_p$ from ppm to Hz; $\alpha_p$ is the decay rate of the corresponding FID signal in the time domain. Chemical shifts and widths, at least for certain peaks, can vary independently and usually reflect the specifics of experimental conditions. For example, the chemical shift of the proton in a hydroxyl group is famously related to the pH value of the sample. On the other hand, relative intensities $b_p$ of peaks pertaining to the same

chemical necessarily remain constant, as they are defined by the atomic composition of the molecule. In the present work, we use the quantum mechanical approach for modelling the signatures of chemical species (Matviychuk et al., 2019). It allows us to minimize the number of free parameters and produce relevant model spectra at any field strength of the spectrometer. Finally, to account for a possibly imperfect baseline in the experimental data, we augment the model matrix $\mathbf{Z}$ with several basis vectors of the form $\mathbf{f}^{l-1}$ for $l = 1, \ldots, L$, which serves to model any polynomial baseline of degree up to $L$ (we use $L = 1$

in all our experiments in Section 4 to correct for the constant offset in the spectra).

With this notation, the complete model spectrum is expressed as $\mathbf{x} = \mathbf{Zc}$, where $\mathbf{c}$ is a vector of component intensities. The main idea of the model-based quantification is to find a model $\mathbf{x}$ that is as close to the measured data as possible; the corresponding vector of intensities $\mathbf{c}$ is used to estimate the concentrations. To formalize this idea, we define the residual spectrum $\mathbf{r} = \mathbf{y} - \mathbf{x}$ and note that $\mathbf{r}$ implicitly depends on the set of model parameters – chemical shifts, peak widths, as well

as the phasing values – which we denote collectively as $\theta$. The model fitting is typically done in the least-squares sense by minimizing the Euclidean norm of the residual:

$$\min_{\theta, \mathbf{c}} \|\mathbf{r}\|_2. \tag{3}$$

It is well known that, given the model matrix associated with the optimal set of model parameters $\widehat{\mathbf{Z}}$, the vector of intensities can be estimated in closed form,

$$\widehat{\mathbf{c}} = \left[\widehat{\mathbf{Z}}^{\mathrm{T}}\widehat{\mathbf{Z}}\right]^{-1}\widehat{\mathbf{Z}}^{\mathrm{T}}\mathbf{y}, \tag{4}$$

where $\widehat{\mathbf{Z}}$ is obtained as a result of unconstrained minimization of the non-linear variable projection functional $\mathcal{L} = \left\|\left(\mathbf{I} - \mathbf{Z}^{\mathrm{T}}\mathbf{Z}\right)\mathbf{y}\right\|^2$.

It can be shown that the criterion of Eq. 3 stems from the assumption that the measured signal is generated as an instance of the model affected by isotropic Gaussian noise, $\mathbf{y} = \mathbf{Zc} + \mathbf{n}$. This plausible assumption is supported by the principle of maximum entropy and the central limit theorem, which made the least squares minimization – along with the existence of a

simple solution – the most popular setting for the model fitting problem. However, as any mathematical model of the physical world, this approach has certain limitations. We discuss them in more detail in the following subsection.

### 2.2   Model misspecification

The optimality conditions of the least squares fit only hold if the assumed signal model is capable of describing the experimental data given some set of parameters. Unfortunately, the most common assumption that underlies model-based qNMR – that an

FID decays mono-exponentially producing spectral peaks with simple shapes – often does not hold in practice. Such effects as diffusion and the magnetic field inhomogeneity cause the resulting peak shapes to deviate from the ideal Lorentzians,





to which a model of Eq. 1 can no longer be perfectly fit. In turn, this leads to incorrect estimates of the intensities of the components **c** and erroneous (biased) quantification results (White, 1981; Deegan Jr., 1976). This stimulated the development of more complex signal models that account for second and higher order effects in the FID, such as Voigt (Marshall et al.,

1997), generalized Lorentzian-Gaussian (Kriesten et al., 2008; Alsmeyer et al., 2004; Schoenberger et al., 2016), or flexible custom lineshapes in numerous spectral deconvolution methods (Cobas and Sýkora, 2009). These approaches were found to be very successful in cases when different peaks in the spectrum, even if they overlap, can be attributed to separate resonances with similar distortions, as often seen in high-resolution data acquired with a high-field instrument. Unfortunately, at the medium field strengths of benchtop instruments, these approaches become less effective. Higher-order coupling between neighboring

and distant protons often cause different $^1$H peaks to show different asymmetric distortions due to separation of transition resonances (Kuprov et al., 2007). Quantum mechanical models were found useful for describing such data but also can not guarantee the perfect fit of complex spectra (Tiainen et al., 2014; Matviychuk et al., 2019).

In this work, instead of trying to refine the peak shape model, which can complicate the analysis and often bears the risk of overfitting, we propose to alter the optimization criterion in order to completely remove any unaccounted signal from the

residual. As an illustrative example, in Fig. 1, we consider a spectrum of thiamine in $D_2O$ acquired on a high-field spectrometer. The high spectral resolution and low level of noise in this dataset make it possible to achieve very accurate quantification results with conventional peak integration. Surprisingly, this appears to be a difficult case for a simple model-based method. The top panel in Fig. 2 demonstrates the least squares fit of Lorentzian peaks to the measured data obtained by minimizing Eq. 3 with respect to the positions and widths of all peaks and the phasing parameters. Close examination of the fit reveals significant

deviations between the experimental and fitted peak shapes. To compensate for the model misspecification, the least-squares fitting distorts the phasing of the spectrum and introduces a notable offset in the baseline. Even though these imperfections are relatively small, less than 1% of the average peak height, they are comparable to the level of random noise and can affect quantification. Furthermore, the mismatch between the model and the data can be easily observed in the residual spectrum: instead of being purely random Gaussian, as postulated in the model assumptions, it is dominated by large spikes where the

model peaks do not fit the data perfectly. The resulting magnitude range of the residual is approximately 100 times higher than the actual noise floor and is at least 20% of the average peak height. Thus, even though the found model admits to the requirements of the optimization criterion, it can not completely explain and account for the measured spectrum. Finally, we note that in this, and many other examples, more flexible peak models (e.g. Lorentzian-Gaussian) still do not eliminate the misspecification error completely.

This observation motivates our proposed approach and distinguishes it from other model-based quantification algorithms: instead of relying on the top-down fitting of a supposedly ideal model, we employ a bottom-up view and aim to find a model spectrum, which after subtraction from the experimental data would lead to exclusively noise in the residual. We present our solution in the following subsection.





**Figure 1.** Examples of spectra of thiamine acquired with a high-field (top) and a medium-field (bottom) spectrometers.

## 2.3 Outline of the proposed adjustment algorithm

The above example demonstrates that the conventional least squares minimization criterion, while being convenient to use, may become inadequate when the assumed model cannot accurately represent the data. The additional useful signal present in



**Figure 2.** Top: An example of a least-squares model fit to a spectrum of thiamine acquired on a 400 MHz spectrometer. Middle: A close-up view of the spectrum. Note that in order to fit Lorentzian peaks to the experimental data, the minimization of Eq. 3 distorts the phasing and forces a constant residual offset in the baseline. Bottom: the residual spectrum after fit; instead containing only random noise, the residual is dominated by large spikes caused by imperfect fit of the lineshapes.

the residual, which was missed by the fitted model, needs to be taken into account when estimating component intensities **c**, especially if absolute quantification is of primary interest.

To develop our solution, we start with the least-squares model fit as described above and represent the residual spectrum **r** as
a sum of three distinct components: a signal remaining solely due to the imperfect model fit that could potentially be accounted for with more flexible signal models, a slow changing residual baseline that arises to compensate for the imperfect fit of the





peaks, and the random noise:

$$\mathbf{r} = \mathbf{y} - \widehat{\mathbf{Z}}\widehat{\mathbf{c}} = \mathbf{r}_m + \mathbf{r}_b + \mathbf{r}_n. \tag{5}$$

Our strategy is to isolate the first term in the above decomposition, $\mathbf{r}_m$, and incorporate it directly into the fitted model, adjusting
the corresponding component intensities $\widehat{\mathbf{c}}$.

We draw the inspiration for our method from the conventional peak integration procedure and note that if the spectrum $\mathbf{y}$ is perfectly phased, its total area under the curve can be found as the sum of all fitted models and the misfit term of the residual, $\mathbf{I} = \sum_{i,k} \left( \widehat{\mathbf{Z}}_{i,k}\widehat{\mathbf{c}}_k + [\mathbf{r}_m]_i \right)$, where the index $i$ runs over all points in the spectrum. In practical applications, where integrals of individual mixture components are of primary interest, the summation is carried out over each column of the signature model
matrix $\mathbf{Z}$ separately, and thus the remainder $\mathbf{r}_m$ needs to be distributed among them, which in turn alters the vector of intensities accordingly. Specifically, we define the resulting component intensities after adjustment as:

$$\widetilde{\mathbf{c}}_k = \frac{1}{\sum_i \widehat{\mathbf{Z}}_{i,k}} \sum_{i=1}^{n} \left( \widehat{\mathbf{Z}}_{i,k}\widehat{\mathbf{c}}_k + \mathbf{W}_{i,k}[\mathbf{r}_m]_i \right), \tag{6}$$

for each $k = 1, \ldots, K$, where $\mathbf{W}$ is an $n \times K$ matrix of row-normalized non-negative weights, $\sum_k \mathbf{W}_{i,k} = 1$, that determine the allocation rule of the residual among the $K$ components at each point in the spectrum $i = 1, \ldots, n$. Note that if the model
is fitted perfectly, and $\mathbf{r}_m = 0$, the normalization $\frac{1}{\sum_i \widehat{\mathbf{Z}}_{i,k}}$ in the adjustment rule of Eq. 6 preserves the original intensities, $\widetilde{\mathbf{c}} = \widehat{\mathbf{c}}$. In our experiments in Section 4, we found it particularly effective to assume that the misfit error of each component is proportional to its value at frequency $i$; then the allocation matrix is defined as:

$$\mathbf{W}_{i,k} = \frac{\widehat{\mathbf{Z}}_{i,k}\widehat{\mathbf{c}}_k}{\sum_k \widehat{\mathbf{Z}}_{i,k}\widehat{\mathbf{c}}_k}. \tag{7}$$

With the assumption that Eq. 6 is capable of recovering the true model intensities, the model adjustment problem reduces
to the isolation of the misfit term $\mathbf{r}_m$ in Eq. 5. For this, we start by explicitly removing the random noise from the residual spectrum, which can be accomplished with any suitable 1D denoising algorithm. We found that soft thresholding of wavelet coefficients is particularly effective for this purpose (Donoho, 1995): it removes the stochastic deviations but preserves the spiky features of the residual that are due to model misspecification. In our examples in Section 4, we use symlets with eight vanishing moments and set universal thresholds proportional to the level-dependent estimates of noise on each wavelet
decomposition level. The resulting signal after denoising, $\mathbf{r}' = \mathcal{D}(\mathbf{r})$, is assumed to be purely deterministic.

Next, we proceed by smoothing the denoised residual to extract its slowly-changing component, $\mathbf{r}_b = \mathcal{S}(\mathcal{D}(\mathbf{r}))$, which encompasses the error introduced by an incomplete baseline correction. While any type of low-pass filtering can be used for this purpose, it is known that median filters – i.e. replacing each point in a signal with the sample median of its $w_m$ neighbours – are especially suitable for removing sharp spike artifacts (Tukey, 1977; Mallows, 1979). To summarize, we
define $\mathbf{r}_m = \mathcal{D}(\mathbf{r}) - \mathcal{S}(\mathcal{D}(\mathbf{r}))$, however, we note that the representation of a residual according to Eq. 5 is inherently an ill-posed problem that does not have a single universal solution; other decomposition strategies can be more effective for different spectra.





As in peak integration, it is paramount in our method that the analysed spectrum $\mathbf{y}$ is perfectly phased before adjusting the models. Fortunately, Eq. 5 provides a convenient way for accurate phase correction, which – unlike most other methods – takes into account information supplied in the form of signature models. We note that the recovered baseline $\mathbf{r}_b = \mathcal{S}(\mathcal{D}(\mathbf{r}))$ implicitly depends on all model parameters and observe that it is particularly sensitive to the linear phasing values, $\varphi_0$ and $\varphi_1$. We demonstrate this with a series of simulations. For this, we generate a spectrum of a single Lorentzian peak with slightly disturbed phase (see Fig. 3); this can, for example, correspond to a residual error after the usual phase correction procedure. We fit this peak with a zero-phase signature model by minimizing Eq. 3 only with respect to the chemical shift and peak width thus intentionally keeping the phase error in the fit. As shown in Fig. 4, the least squares fit of an imperfectly phased peak leads to erroneous estimates of its position and width. Consequently, phase error of approximately 0.4 rad is able to cause an error in the peak intensity estimate of about 5%. Thus, it is important to eliminate any phasing imperfections if the desired accuracy of quantification lies below 5%.

Although there is no random noise added to the spectrum in the above example, the phasing error manifests itself in the residual in Fig. 3. Notably, the asymmetry of an imperfectly phased peak induces positive and negative tails in the residual; this effect is greatly emphasized by median filtering, which creates a sharp transition in $\mathbf{r}_b$. Naturally, we desire to recover the residual baseline as smooth as possible, and thus penalising such sharp edges is an especially effective strategy for fine-tuning of the phasing parameters. In this work, we found that the same criterion of Eq. 3 but now applied only to the extracted residual baseline works most effectively for this purpose, i.e. to adjust the phasing, we minimize

$$\min_{\varphi_0, \varphi_1} \|\mathbf{r}_b\|_2. \tag{8}$$

Furthermore, multistage median filtering with increasing window widths $w_m$ tends to improve the smoothing results, which agrees with recent works (Arias-Castro and Donoho, 2009). Intuitively, $w_m = 0$ corresponds to no smoothing, and the problem of Eq. 8 reduces to the original least squares formulation of model fitting (Eq. 3), except for the removed noise. On the other hand, very broad filters on the order of the total spectral width tend to produce smooth results; in turn, this makes them ineffective for the above optimization, whose goal is to remove sharp edges by adjusting the phase. Notably, in our simulations, the minimum is attained at the true value of the phasing parameter $\varphi_0$ using median filters at least eight times wider than the width of the peak (see Fig. 5). Thus, we propose to solve the problem of Eq. 8 iteratively: given an average peak width in the spectrum, FWHM, we start with a median filter of size at least $w_m > 10 \cdot \text{FWHM}$, minimize Eq. 8 with respect to the phasing parameters, and then increase $w_m$ to recover a smooth residual baseline $\mathbf{r}_b$ at the final optimization stage.

We apply the same method to adjust both phasing parameters in the experimental data of thiamine in $D_2O$. Fig. 6 displays the values of the cost function in Eq. 8 plotted with respect to the deviation in the phasing parameters $\varphi_0$ and $\varphi_1$ from their current values. We note that the phasing of the spectrum after the usual least squares fit with Eq. 3 is suboptimal in the sense of our criterion based on the smoothness of the baseline. The proposed adjustment reduces the fitting cost (note the lower minima after adjustment, especially for $\varphi_0$) and significantly improves the phasing of the resulting spectrum as shown in Fig. 7. Furthermore, Fig. 8 displays the residual baseline before and after minimizing Eq. 8 computed using median filters of two different sizes. Note the conspicuous sharp transition present in $\mathbf{r}_b$ after the least squares fit that is due to imperfect





**Figure 3.** A simulated example of fitting an imperfectly phased peak using a zero-phase model with adjustable position and width. A phasing error of 0.25 rad shown here causes incorrect estimation of the chemical shift and creates sharp spikes in the residual spectrum $\mathbf{r} = \mathbf{y} - \mathbf{x}$ (middle). The baseline remaining after the fit, $\mathbf{r}_b$, is found by smoothing $\mathbf{r}$ using median filters with window width $w_m$ defined relative to the peak width at half maximum (FWHM), $w_m = k \cdot \text{FWHM}$ (bottom).





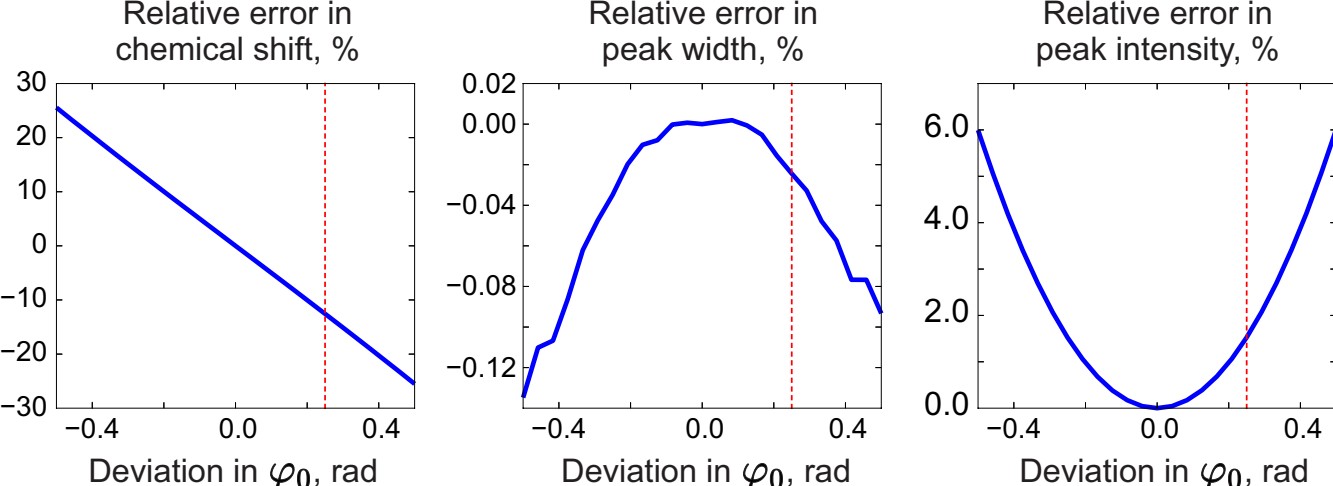

**Figure 4.** Relative errors in estimated model parameters fitted to an imperfectly phased peak as functions of the phasing error. From left to right: error in the chemical shift (expressed relative to the peak width at half maximum), error in the peak width, and error in the intensity relative to their respective true values. The red vertical lines indicate the phasing error of 0.25 rad corresponding to the plots in Fig. 3.

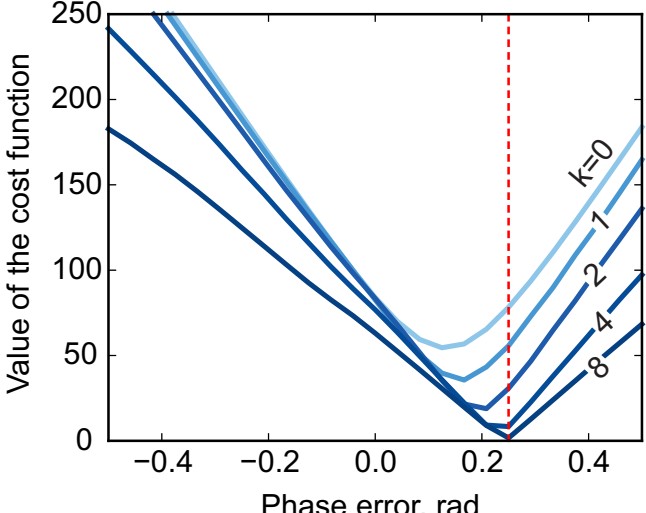

**Figure 5.** The phase-adjustment cost function, $\|\mathbf{r}_b\|_2$, plotted against the phasing error. Different curves correspond to median filters of different width, $w_m = k \cdot \text{FWHM}$. The red vertical line indicates the true phasing offset in the simulations.

phasing. The proposed phase adjustment reduces peak-to-peak deviations in the residual baseline $\mathbf{r}_b$ by more than four times, and further filtering with a wider window produces an almost flat baseline, which does not exceed the natural level of noise, as desired. This recovered baseline is now safely removed from the phased spectrum without affecting the areas under the peaks.

Unlike the least-squares criterion of Eq. 3, smoothing the residual baseline with Eq. 8 does not directly penalize the misfit between the model and the data; in general it leads to slightly higher mean-squared error between $\mathbf{x}$ and $\mathbf{y}$. However, it allows us to isolate the remaining unaccounted signal $\mathbf{r}_m$ in the residual from the baseline and noise, which would have been naturally





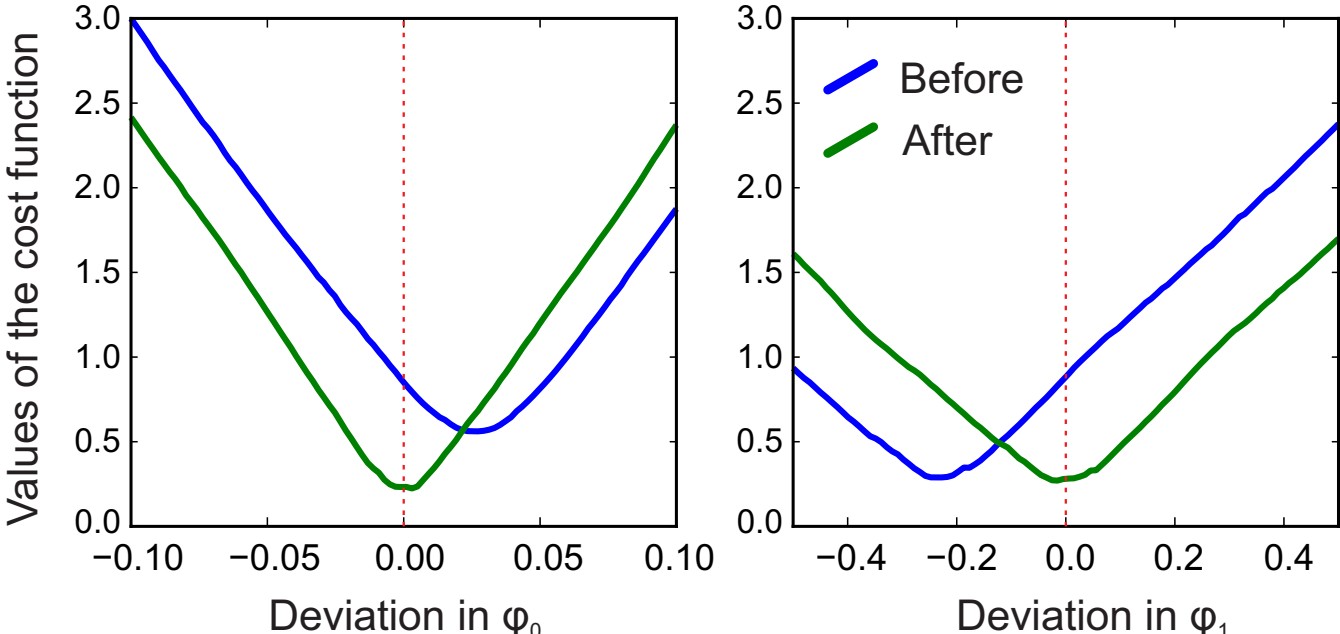

**Figure 6.** Phase adjustment of the experimental spectrum of thiamine. The norm of the residual baseline $\|\mathbf{r}_b\|$ as a function of deviation in the parameters of the linear phasing model from their current values (0.0 on the horizontal axes), $\varphi_0$ (left) and $\varphi_1$ (right) before and after their adjustment. The optimal phasing parameters estimated with the initial least squares fit using Eq. 3 (blue lines) are suboptimal in the sense of the adjustment criterion of Eq. 8. Median filter of size $w_m \approx 100 \cdot \mathrm{FWHM}$ is used in this example.

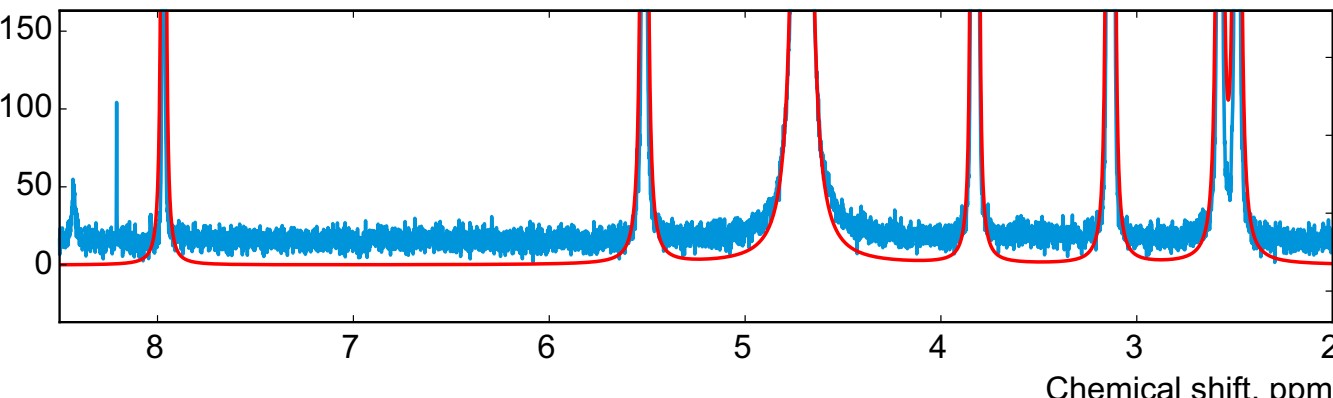

**Figure 7.** The spectrum of thiamine after phase adjustment according to the proposed rule (cf. Fig. 2).

included in peak integration and distribute it among the model components. This eventually results in more accurate estimation of their intensities, with minimal additional computational effort, as we demonstrate in Section 4.





**Figure 8.** The residual signals before (top) and after (middle) adjusting the phase parameters using a median filter with size $m_w \approx 100 \cdot$ FWHM. Bottom: The result of applying a wider filter with $m_w \approx 400 \cdot$ FWHM.

## 3   Materials and Methods

### 3.1   Sample Preparation and Data Acquisition

For the first part of our experiments, we prepare a sample of $0.5$M thiamine dissolved in $D_2O$. Thiamine hydrochloride was purchased from Sigma Aldrich and has purity of $99.0\%$ by weight specified by the manufacturer. The measurements were performed on a $400$ MHz Agilent 400MR spectrometer equipped with a OneNMR probe. We acquired $16384$ time points with dwell time of $156\,\mu$s and pulse angle of $45°$ with a single scan. This results in average SNR of approximately $2000$ for the peaks of thiamine.





For the second set of experiments, we prepare 22 samples of organic mixtures in varying relative concentrations (please see supplemental material for more detail). Methanol and Ethanol were purchased from Merck KGaA and have purity specified by the manufacturer 99.8% and 99.9% by weight respectively. Methyl acetate and ethyl acetate were purchased from Sigma
Aldrich; the purity of both species is 99.8% by weight.

We use a Mettler Toledo AX205 balance with instrument accuracy of 0.1 mg (provided in the calibration protocol of the manufacturer). By means of the accuracy of the laboratory balance and error propagation, the uncertainty of the gravimetrically determined mole fraction was estimated to be $1.29 \cdot 10^{-5}$ mol/mol.

In this experiment, the data were acquired on a high-field NMR spectrometer with a 9.4 T vertical superconducting magnet
(Ascend 400, console: Avance 3 HD 400, Bruker Biospin, Rheinstetten, Germany), which correspond to a proton Larmor frequency of 400.13 MHz equipped with a standard probe (BBFO, Bruker Biospin, Rheinstetten, Germany). We use proton NMR experiments and a simple one-pulse sequence with a pulse angle of $30°$ and $^{13}$C inverse gated decoupling. For each sample, we collect 20028 points with a dwell time of $250\,\mu$s, and repeat the acquisition with 16 scans and a relaxation delay of 30 s. The instrument was tuned and shimmed individually for each sample. For processing, the datasets were extended to $2^{16}$
points by zero-filling. The SNR in these datasets was estimated to be $100 - 10^4$ depending on the specific peak considered.

Additionally, the same samples were measured with two medium-field benchtop spectrometers, Magritek Spinsolve (for the thiamine sample) and Magritek Spinsolve-Carbon (for the organic mixtures). These spectrometers operate at a $^1$H frequency of 43.13 and 42.63 MHz, respectively. In $^1$H experiments, we collected $2^{15}$ time points with a dwell time DW $= 200\,\mu$s. The experiments were run with single scans and the pulse angle of $90°$. While collecting the data, both Spinsolve instruments were
periodically recalibrated using the standard shimming protocol to ensure the best field homogeneity.

### 3.2    Data Processing and Quantification

Peak integration and qGSD analysis were carried out with the software Mnova (version 14.0.1, Mestrelab Research, Santiago de Compostela, Spain). In each case, automatic phase (global, whitening) and baseline (Whittaker smoother or polynomial fit of the third degree) corrections were applied followed by visual inspection and manual adjustment where necessary. Integration
boundaries for each peak are chosen based on their full width at half-maximum (FWHM) and are set at least $50 \times$ FWHM. Quantitative GSD was run with manual range selection and five improvement cycles.

The least-squares fitting and the proposed adjustment algorithm were implemented in a custom software written in Python 3.5.

For each sample $s$, we report the results of quantification with all methods in terms of the root mean square error ($\mathrm{RMSE}_s$) in mole fractions computed with respect to the values obtained gravimetrically, $x_{s,k}^{grav}$:

$$\mathrm{RMSE}_s = \sqrt{\frac{1}{K}\sum_{k=1}^{K}\left(x_{s,k}^{est} - x_{s,k}^{grav}\right)^2},$$

where $x_{s,k}^{est}$ is the mole fraction of the $k^{\mathrm{th}}$ species estimated in the $s^{\mathrm{th}}$ sample and expressed in mol/mol. The average RMSE is computed over all $S$ samples, $\mathrm{RMSE}_{avg} = \frac{1}{S}\sum_{s=1}^{S}\mathrm{RMSE}_s$.





## 4    Results and Discussion

In this section, we apply the proposed adjustment procedure for model-based quantification to two sets of samples. First, we
study the performance of our algorithm using a sample of thiamine in $D_2O$ and compare the relative ratios of its peaks with
the known ground truth. In the second example, we analyse a set of organic mixtures prepared gravimetrically. In both cases,
we look at data acquired with a high-field spectrometer as well as a medium-field benchtop instrument.

### 4.1    A Sample of Thiamine in $D_2O$

For the first series of experiments, we prepare a sample of 0.5 M thiamine dissolved in $D_2O$, which we referred to previously
in Section 2. We choose this compound for its very characteristic $^1$H NMR spectrum: it exhibits a pair of well-separated peaks
at 5.45 and 7.9 ppm, a pair of partially overlapping peaks at 2.42 and 2.52 ppm, and a pair of triplets at 3.07 and 3.77 ppm
(see Fig. 1). This allows us to test various aspects of our model, including the quantum mechanical formulation of coupled
spin systems (see Matviychuk et al. (2019) for more detail). Since the true ratios of peak intensities are known and fixed,
by estimating them separately – as if the peaks belonged to several chemical species in unknown concentrations – we can
unequivocally compare different qNMR methods in terms of their accuracy. In these examples, we refer to and estimate the
intensity of the three peaks that comprise each triplet collectively.

We consider a spectrum acquired with a high field spectrometer, and also analyse the same sample with a benchtop system.
In the former case, the conventional peak integration readily achieves errors in relative mole fractions as low as 0.001 (see
Fig. 9). Furthermore, quantitative global spectrum deconvolution (qGSD) performs excellently in this example and shows
significant improvement compared to the standard GSD algorithm (Cobas et al., 2008; Cobas and Sýkora, 2009). The ratios of
peak intensities estimated with the least squares model fitting (LS) are also more accurate than the GSD results. However, due
to inevitable lineshape misspecifications, the LS method can not outperform the peak integration of well separated peaks in a
low noise spectrum as well as qGSD, which relies on a sophisticated deconvolution of each peak individually. On the other
hand, the proposed adjustment algorithm significantly improves the LS results and brings the root mean-squared-error (RMSE)
to the level achieved with peak integration (see Fig. 10).

It is instructive to look at the possible improvement of the least squares fitting results with more representative lineshape
models. Specifically, we include second and third order terms in the real and complex-valued decay model of FID as done
in (Matviychuk et al., 2017), which contribute additional weighting parameters to be fitted; we observe that the second order
FIDs correspond to linear combinations of Lorentzian and Gaussian lines in the spectrum, while complex-valued polynomial
decay models allow us to address peaks asymmetry. Furthermore, we consider a custom-written version of the reference
deconvolution method (Metz et al., 2000), in which we estimate the convolution kernel given a Lorentzian model fitted to
the experimental data as described above. This method has the highest potential to represent various lineshape deviations
but nevertheless is restricted by using the same convolution kernel for all peaks. The quantification results in these cases are
summarized in Fig. 11 in terms of the RMS errors in peak ratios as well as the values of fitting objective of Eq. 3. As expected,
more complex signal models allow us to fit the experimental data better and reduce the norm of the residual **r**. However, better

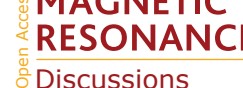

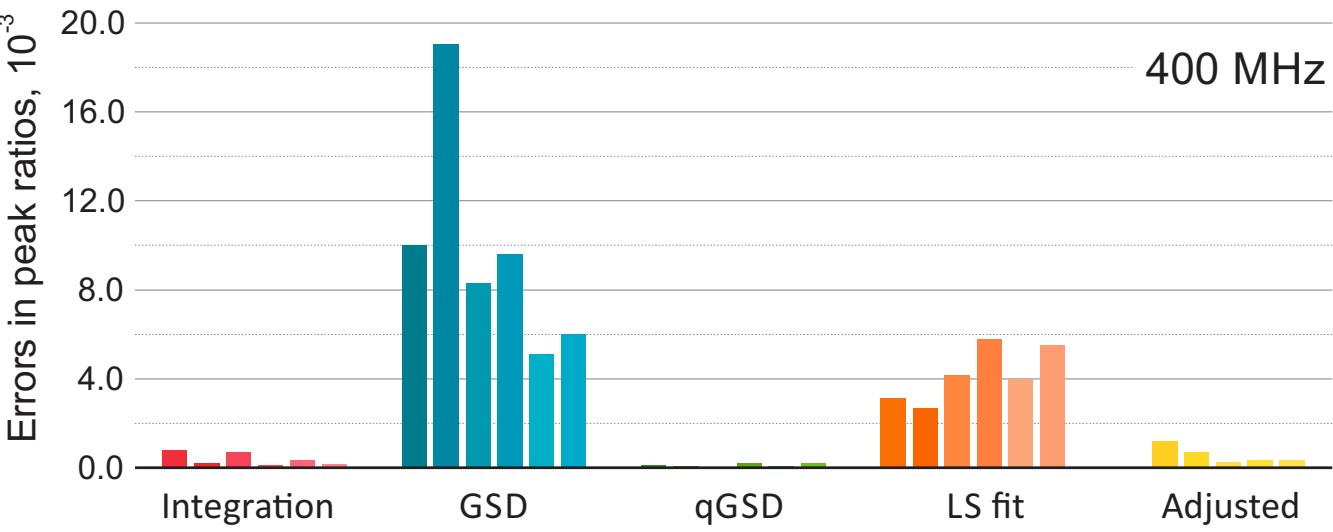

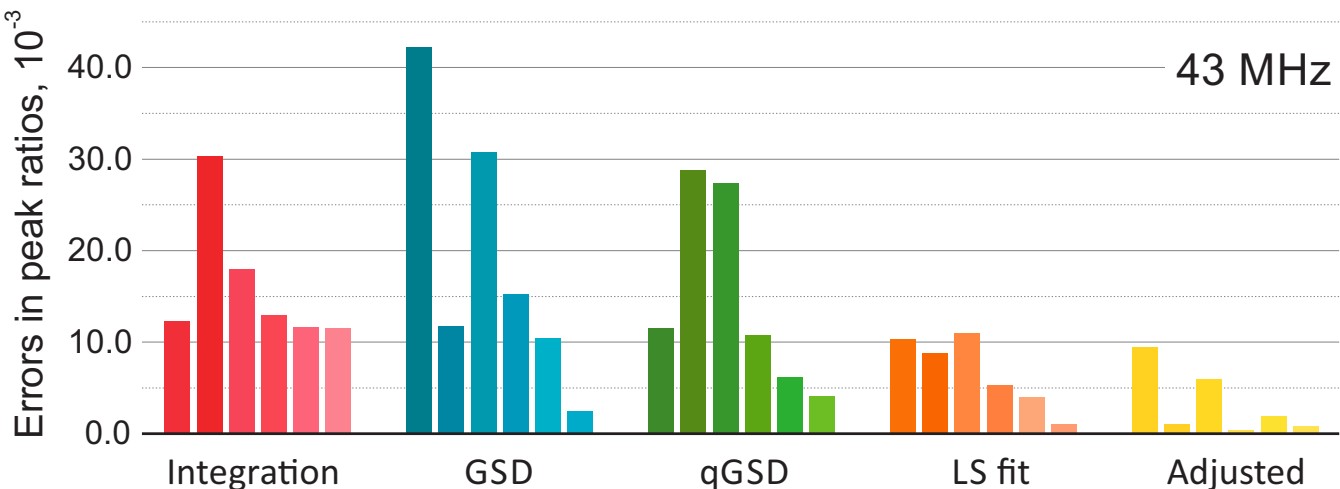

**Figure 9.** The results of quantitative analysis of thiamine spectra acquired on a high field (top) and medium field (bottom) spectrometers. The plots show absolute errors in ratios of peak intensities estimated with five different methods: the traditional peak integration, global spectral deconvolution (GSD) and its quantitative modification (qGSD), least-squares model fitting (LS), and the proposed adjustment algorithm applied to the LS fit. In each group, the bars correspond to the peaks of thiamine ordered from left to right according to their chemical shift. Note that the adjustment method improves the accuracy of model-based quantification for all peaks in both cases.

LS fit does not always entail lower quantification errors, which signifies possible overfitting. On the other hand, the proposed phase adjustment by minimizing $\|\mathbf{r}_b\|$ leads to slight increase of the total residual norm $\|\mathbf{r}\|$, but the following distribution of $\mathbf{r}_m$ among the reference signals according to Eq. 5 significantly reduces the quantification error.





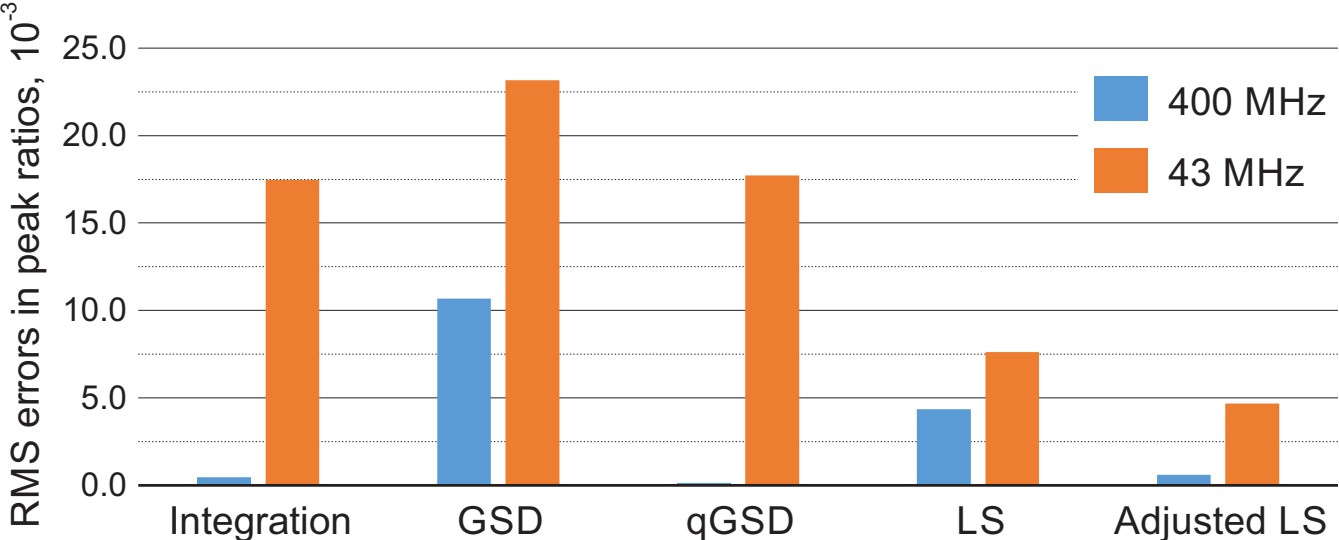

**Figure 10.** Root mean square errors (RMSE) of estimating peak ratios in spectra of thiamine computed by averaging the results in Fig. 9. The peak integration and the qGSD algorithm are very effective for the analysis of well resolved high-field data. The proposed adjustment method improves the LS quantification results even when peak overlap is present, such as in the medium-field spectra.

Analysis of the same sample acquired on a benchtop spectrometer is a more challenging task for the established methods.

Partially overlapping methyl peaks at 2.5 ppm make it difficult to define ranges for their integration. On the other hand, GSD methods, that lack information about the underlying molecular system, often tend to define a third broad peak that overlaps with these two main resonances to compensate for lineshape broadening near the baseline. Although this produces a plausible fit overall, the spurious extra peak does not have any physical meaning and is difficult to account for in the final quantification results (we attribute its area to the closest resonance in our analysis). The distortions in lineshape of the triplets at 3-4 ppm are

caused by higher-order coupling effects rather then a magnetic field inhomogeneity, and thus modelling them with a quantum mechanical approach is especially effective here. Even though the RMSE of the LS model fit is more than two times lower than that of peak integration and qGSD, the proposed adjustment method allows us to further reduce it by almost 40%. The remaining error is mostly due to one of the overlapping methyl peaks, whose intensity estimate is particularly sensitive to small deviations in phasing parameters. As in the high-field example, the bottom panel in Fig. 11 shows the quantification accuracy

achieved with alternative lineshape models. The proposed phase adjustment method with the distribution of the residual has lower quantification error that the reference deconvolution approach and does not require fitting any additional lineshape parameters, unlike the the higher-order peak models.

## 4.2 A set of Organic Mixtures

Next, we study a set of 22 mixtures of methanol, ethanol, methyl acetate, and ethyl acetate prepared gravimetrically in varying

relative concentrations ranging from 0.02 to 0.95 mol/mol for each component; as before, we measure their spectra with a

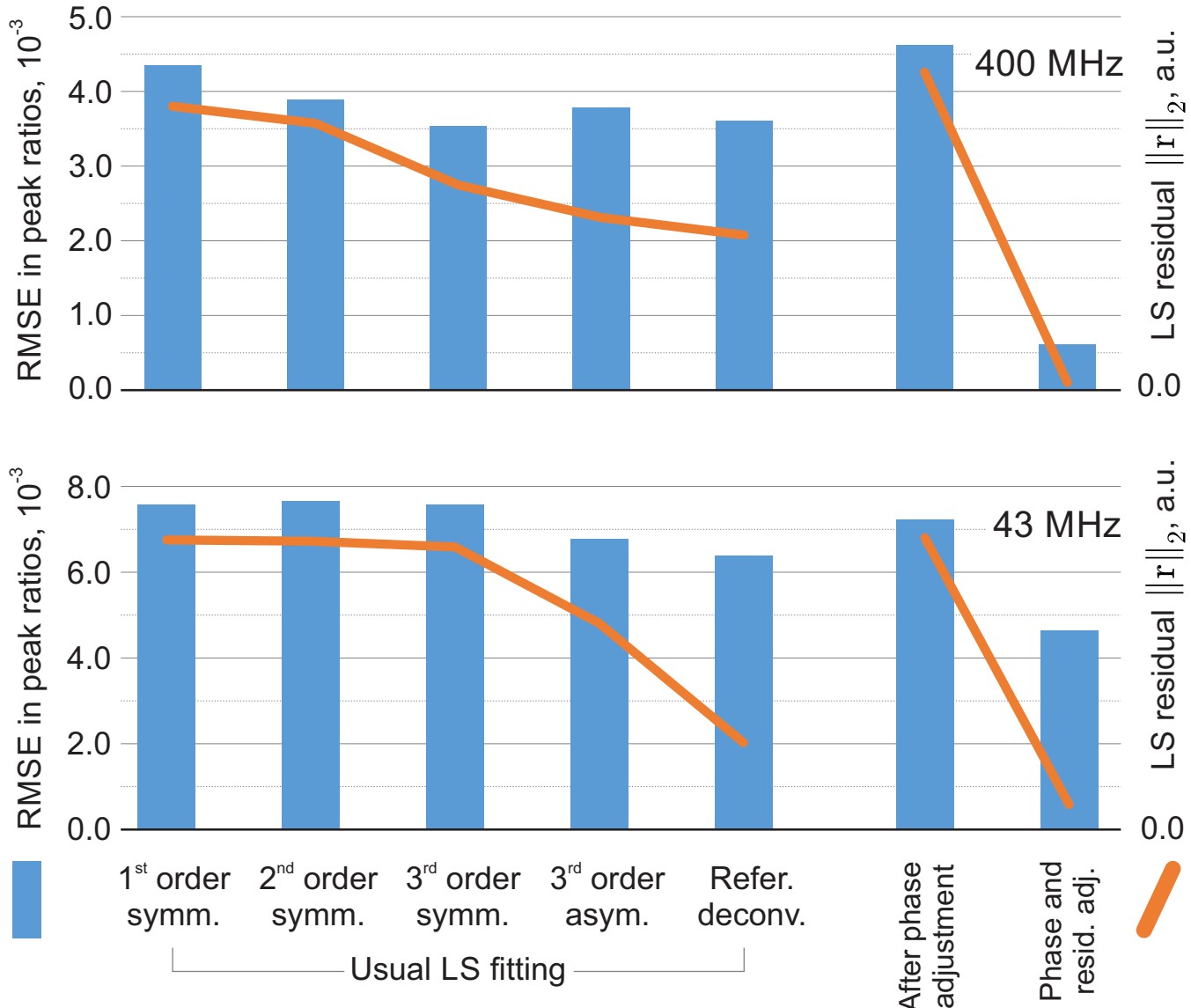

**Figure 11.** Results of LS fitting of high field (top) and medium field (bottom) data using models that take into account higher order lineshape distortions. From left to right: first order Lorenzian, second order symmetric model (weighted combination of Lorentzian and Gaussian lineshapes), third order symmetric and assymetric peak models. In the reference deconvolution approach, a single kernel is estimated for all peaks based on the difference between the experimental spectrum and a fitted Lorentzian model. For the proposed method, the results are reported after the phase adjustment (Eq. 8) and the final distribution of the residual (Eq. 6). Note that lower norms of the residual achieved by fitting more flexible models to the data do not always entail reduced quantification errors, whereas the proposed adjustment method achieves this goal more efficiently.





high-field and a medium-field benchtop spectrometer (see Fig. 12). Using these datasets, we estimate the mole fractions of each chemical species in the mixtures and compare them with the gravimetric values. To define the models for all chemical species, we use their complete quantum mechanical formulations and find the corresponding chemical shifts and J-couplings using the high-field data (Matviychuk et al., 2019). We use these parameters to initialise corresponding models at the lower field strength of the benchtop instrument and then refine them by minimizing Eq. 3.

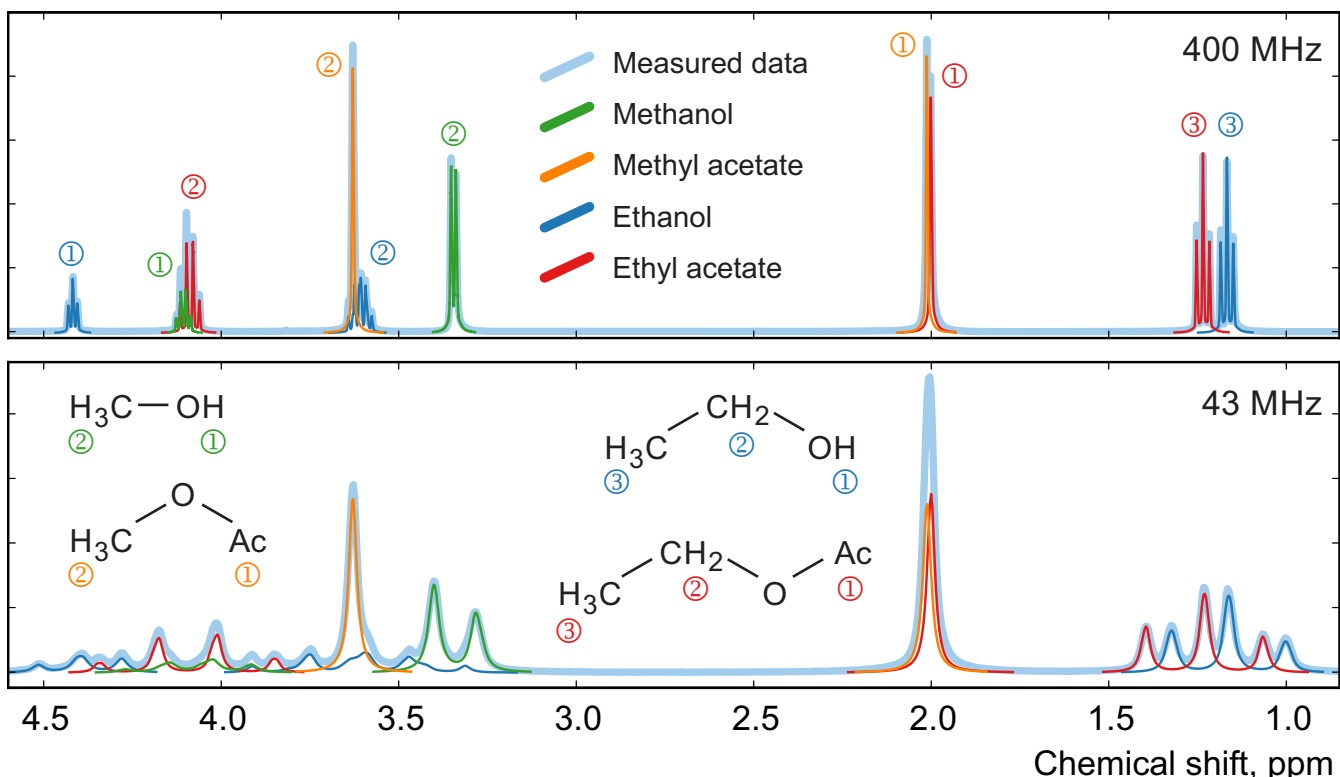

**Figure 12.** Examples of spectra of a mixture of four organic compounds acquired on a high-field (top) and medium-field (bottom) spectrometers. In this sample, the mole fractions of all four components are approximately equal. Numbers next to the peaks indicate their assignment to specific [1]H atoms in the studied molecules.

For comparison, we apply the qGSD algorithm to the high-field datasets: specifically, we include all non-overlapping peaks in our analysis and also peaks of proton in the acetate groups (see two overlapping peaks at 2.0 ppm in Fig. 12). Where possible, we also take into account the peaks of protons in the hydroxyl groups. On the other hand, severe peak overlap in the benchtop spectra precludes their accurate assignments and deconvolution, which makes this dataset extremely challenging to analyse with the traditional methods; therefore, methods based on fitting of quantum mechanical models are especially useful in this example. The RMSE results of quantification of eight representative samples along with the average values across all 22 samples are shown in Fig. 13; detailed results of the complete analysis of this dataset can be found in supplementary material. With the high field data, the least squares model achieves accuracy of quantification similar or slightly better than the qGSD





algorithm, and the proposed adjustment procedure reduces the average error in mole fractions by almost $50\%$. As expected,

the analysis of the benchtop data results in slightly higher errors in mole fractions – with RMSE of up to $0.04$ mol/mol for

certain samples, which nevertheless is acceptable in many practical applications. However, the average error across all samples,

$\mathrm{RMSE}_{avg}$, is comparable to that achieved by the model-based methods (qGSD and LS) with the high field dataset, and the

proposed model adjustment further reduces it by $25\%$ on average. Finally, we note that occasionally – especially with the

benchtop data – the adjustment of the LS fit results in slightly higher quantification errors (e.g. see the results for the seventh

sample in Fig. 13). The increase in the error is likely due to an imperfect mechanism of distributing the residual among the

overlapping signature models. The assumption that the error is proportional to the component intensity as postulated in Eq. 7

may appear insufficient in these cases; the development of more accurate allocation rules is a topic of ongoing work. However,

the increase is usually less than $0.005$ parts in mole fractions, and if the entire data set is considered the average error is reduced,

as already noted.

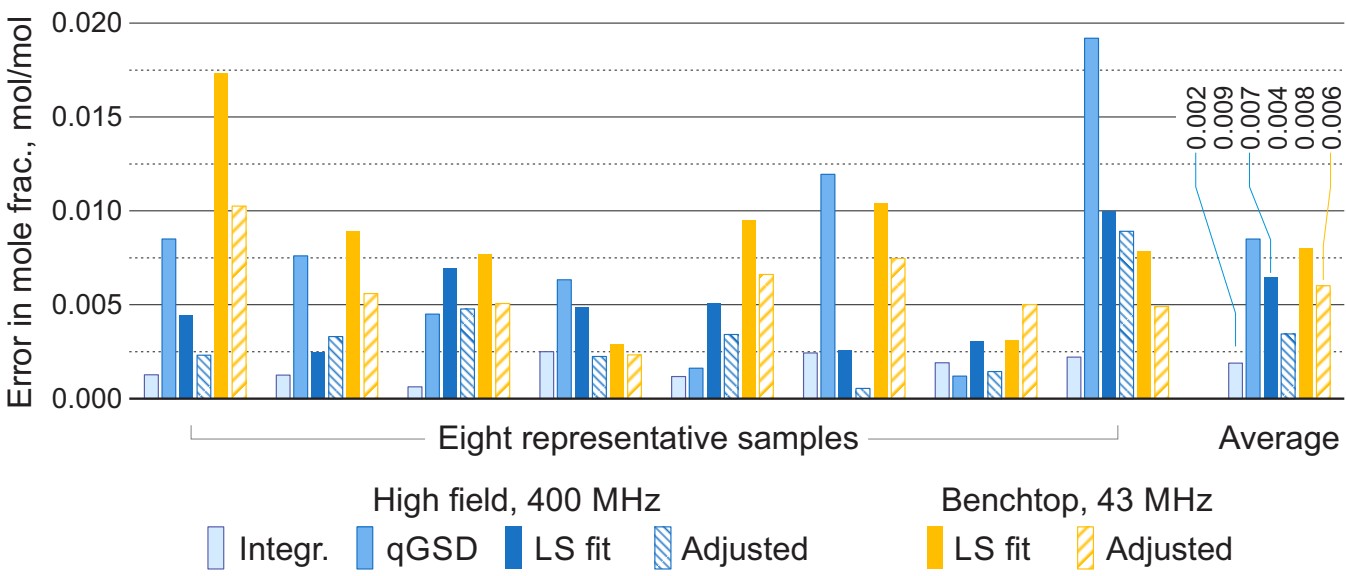

**Figure 13.** Root mean square errors in estimated mole fractions with respect to gravimetric values in selected representative samples from the set of organic mixtures. The proposed adjustment almost always improves the LS estimates and brings the accuracy of benchtop results to the level comparable with qGSD analysis of high-field data. The average errors are computed over all 22 samples.

## 5    Conclusions

We proposed an effective and computationally simple mechanism to improve the accuracy of model-based quantification in NMR data analysis. The proposed adjustment procedure aims to account for all useful signal left in the residual after the usual least squares fit, which can signify a case of model misspecification – a problem notoriously difficult to avoid in most model-based qNMR methods. Our alternative optimization criterion explicitly relies on the denoising of residual and smoothing the



remaining baseline and is particularly effective in correcting errors in spectrum phasing. The results of analysis of experimental datasets obtained with high and medium field spectrometers indicate the accuracy improvement by 20-40% compared to the usual least-squares model fit.

*Data availability.* NMR spectra in the JCAMP-DX format are available as supplementary material.

*Author contributions.* DH and EH conceived the project. All authors designed the experiments. ES, DH, and YM conducted the experi-
ments and analysed the data. YM developed the computational algorithm and drafted the manuscript. All authors reviewed and edited the manuscript.

*Competing interests.* The authors declare that they have no competing interests.

*Acknowledgements.* Financial support of the present study by the New Zealand Ministry of Business, Innovation and Employment (grant number UOCX1502) and Deutsche Forschungsgemeinschaft DFG (project number 310714510) is gratefully acknowledged.





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
