# Peer review of "Improving the Accuracy of Model-based Quantitative NMR"

_Magnetic Resonance, 2019_

## Referee Comment (RC1) · Anonymous Referee #1 · 9 Feb 2020

The manuscript describes an alternative optimization routine (aka model adjustment procedure) to improve the accuracy of the model-based qNMR. The problem with existing frequency domain analysis is well described and the method proposed seems to certainly improve existing frequency domain methodology (GSD and its variants).

I wonder if they have suggestion and/or examples of such improvements with the existing time-domain modeling approach. While they eluded to such potential improvement in the introduction section for time-domain approach as well, there are no results to back that. I have particularly two concerns.

1. Fundamentally the Bayesian approach of Bretthorst (which is the engine behind the CRAFT method) does not fit each resonance to one given model (typically exponentially decaying sinusoid). Rather it fits the supplied time-domain signal (FID or digitally

filtered subFID) to as many exponential decay models as needed to completely define it. I completely agree that a resonance (as per the NMR definition), more often than not, is non-Lorentzian in shape (for various reasons as pointed out by the authors). Attempting to fit that with a single model type (Lorentzian or gaussian or whatever the fancy maybe) will almost certainly bound to leave behind some part of the signa. The iterative approach of Bayesian approach defines such a resonance with "as many exponential models" as needed (again, more often than not) to completely define it (of course, within the limits of the residual noise and the probability function). The CRAFT approach, in particular, does not equate (or approximate) one resonance to one "best" exponential model, but rather as a complex sum of all the exponential models within a "segment width" (defined and introduced in the reference MRC 51, 821 and used in other subsequent reports) that defines the resonance (aka fingerprint). Such "segment definition" is best done as post data-decimation rather than as a constraint to it. CRAFT approach, hence, circumvents the such potential non-Lorentzian modeling by multi-exponential sinusoidal decay function. In other words, conceptually a resonance is not defined by the "best singular" model but rather by as many models needed to define it completely. 2. In time-domain method-optimization using residual spectrum (FT of the residual FID) as can be misleading – particularly when dispersive residuals are used as a guide to the efficiency of amplitude estimation. Majority of the cases (in time-domain analysis) the dispersive residuals are the result of error in frequency estimation. For example, for a resonance of 1 Hz linewidth an error in frequency estimation by 0.02% (100 Hz vs 100.02 Hz - in chemical shift scale on a nominal 400-500 MHz spectrometer, this is an error of 20 ppt – parts per trillion!) leads to recognizable dispersive residual in the spectrum – visualized after FT of the residual FID. Considering frequency and amplitude are orthogonal parameters in defining the NMR signal, improvement attempts to correct for such small error in frequency estimation to achieve "perfect" amplitude may be an exercise of academic interest.

I do agree with the authors and their methodology in improving qNMR estimation by frequency domain modeling approach. But it would be appropriate to put a note of

caution that such expectations of improvements are yet to be realized for time-domain methods. Alternately, I will like to see some results if similar improvements are to be extrapolated to time-domain modeling approach. Else, it could be prematurely misleading.

---

## Referee Comment (RC2) · Anonymous Referee #2 · 25 Mar 2020

The authors propose an improved algorithm for quantitative NMR (qNMR) through model adjustments. The introduction section provides a good review of a number of challenges that are commonly faced in model specification. Although, a full discussion related to the specification of number of components in the model – a factor that can contribute significantly to model misspecification – is not offered. Overall, the discussion in the Introduction section is well-presented.

There is an example involving the modeling of a mixture in the manuscript which leads me to think that deconvolving mixtures is a supported feature. If this is correct, then perhaps the authors implicitly assume that the number of components are a known quantity? If so, then this assumption should be stated explicitly since the number of components in a linear mixture model in the presence of noise can have influence the

calculated parameters of the model in a significant way.

On the other hand, if the authors assume that the number of components is an unknown parameter, then two additional points for discussion should be included in order to help readers develop a fuller picture: a) what procedures should be used for deriving the number of components?, and b) what would be the impact of the number components on the "model adjustment" process proposed by the authors?

It is worth noting that NMR signals are non-stationary, and there is supporting data for the view that the appropriate noise model for NMR spectra is not Gaussian. As a result, the outcome of the model fitting stage, depending on the objective function used for optimization, may lead to a residual noise background that is challenging to characterize. More specifically, the underlying distribution of this residual noise may change (according to the standard metric) in every iteration of an algorithmic optimization and at every point of the spectrum. This phenomenon, combined with lack of a priori knowledge about the number of components in the model, can hinder the adjustment procedure for weaker peaks.

As stated in the manuscript, the model adjustment process attempts to "... find a model spectrum, which after the subtraction from the experimental data would lead to exclusively noise in the residual." This is an intuitively appealing objective, but only a heuristic (not rigorous) form of this statement is given. The authors state in a later paragraph that the specification of this objective through equation 5 is ill-posed, but their statement does not explicitly address the issue of how "exclusively noise" can be determined. Recall that two noise vectors of identical power (or different power) can be far apart in the standard metric but rather close in an appropriately selected metric. For example, the cosine distance for two noise vectors generated from the same random process can be close to 1 ($1 - r$, where $r$ is the cosine between two noise vectors, is considered very far). Therefore, the precise specification of how the residual becomes exclusively noise is necessary.

[Figure]

**MRD**

The authors address the challenge of addressing "noise equality" by focusing on the component of the residual signal that is related to the model "misfitting" in the decomposition presented in equation 5. Their proposal is to remove the noise through the use of symlet wavelets and a set of filters and consider the residual as "misfitting". Since symlets are dependent on the order of their filter, and there are a multitude of filtering approaches, the proposed approach has the side effect of introducing additional parameters. The authors present a figure showing the impact of symlet order on RMSE in peak ratios which seem to indicate the superiority of the symlet over reference deconvolution. However, the RMES in peak ratios seem to suggest very small improvements as a result of applying the proposed algorithm. Is this correct? There are no quantitative guidelines for the selection of these parameters and the implicit assumption that the reader is expected to arrive at based on the examples presented is that the impact of parameters is negligible. Is this the claim that is being made?

As a tool for user-supervised spectral fitting, the proposed approach appears to be useful, assuming that the software in source form is provided to the community. Do the authors envision the use of this software as a user-supervised tool? Or, do they consider the software as providing an automated procedure? There is plenty of computational power in a typical desktop to perform near real-time changes and visual feedback based on user-adjustable parameters, if a friendly GUI for the task is provided. Perhaps this GUI could be something developed by the community if one had not been developed already. It is unclear if this material is available and what licensing rights are available for future open developments.

Some additional discussions will add significantly to the science presented in the manuscript: a) Additional clarifications regarding the comments/questions mentioned above. b) Clear and concise statements regarding the assumptions made in devising this algorithm. For example, is it assumed that the number of components is known a priori? c) The capabilities and limitations of the approach. For example, how many peaks can be handled and how much overlap is tolerated by the algorithm? d) modes

of parameters specification in the software, and guidance on the method of comparing outcomes will add. If it is being used as a user-guided tool, what utilities are available for guiding the user in selecting the "best" solution? If it is left to the user, then please state it. e) the use of software as a user-supervised tool. Do the authors view this as an automated tool, or is it viewed as a user-operated tool to gain insight? f) Availability of the software and license rights.

---

## Author Comment (AC1) · 7 Apr 2020

**Comments from the Editor**

Citation of relevant literature on model-based analysis of NMR spectra appears to be inadequate. In particular, the authors may wish to explore prior work from Bretthorst (Bayesian) and Markley (Maximum Likelihood).

[Figure]

Response

Thank you for this suggestion. Additional citations of foundational works have been added to the Introduction (Miller and Greene, 1989; Bretthorst, 1990; Chylla and Markley, 1995).

**Comments from the First Reviewer**

The First Reviewer has brought up an important question of the possibility to extend our method to time-domain model-based qNMR. Even though this falls out of scope of the current paper and no work has been done in this direction, we recognize the importance and the practical need for such extension. Specifically there have been two concerns raised.

Comment 1

Fundamentally the Bayesian approach of Bretthorst (which is the engine behind the CRAFT method) does not fit each resonance to one given model (typically exponentially decaying sinusoid). Rather it fits the supplied time-domain signal (FID or digitally filtered subFID) to as many exponential decay models as needed to completely define it. I completely agree that a resonance (as per the NMR definition), more often than not, is non-Lorentzian in shape (for various reasons as pointed out by the authors). Attempting to fit that with a single model type (Lorentzian or gaussian or whatever the fancy maybe) will almost certainly bound to leave behind some part of the signa. The iterative approach of Bayesian approach defines such a resonance with "as many exponential models" as needed (again, more often than not) to completely define it (of course, within the limits of the residual noise and the probability function). The CRAFT

approach, in particular, does not equate (or approximate) one resonance to one "best" exponential model, but rather as a complex sum of all the exponential models within a "segment width" (defined and introduced in the reference MRC 51, 821 and used in other subsequent reports) that defines the resonance (aka fingerprint). Such "segment definition" is best done as post data-decimation rather than as a constraint to it. CRAFT approach, hence, circumvents such potential non-Lorentzian modeling by multi-exponential sinusoidal decay function. In other words, conceptually a resonance is not defined by the "best singular" model but rather by as many models needed to define it completely.

Response

The CRAFT method that uses the Bayesian machinery of Bretthorst is indeed very successful in representing non-ideal lineshapes in the spectrum as complex sums of exponentially decaying sinusoids in the time-domain. Unfortunately, this raises a challenging problem of assigning the entries of the resulting frequency-intensity table to individual chemical species components of the analysed mixture. We have added a note in this regard to the manuscript in the Introduction page 2: "Alternatively, CRAFT [Krishnamurthy2013], the popular time-domain method based on the iterative Bayesian machinery of Bretthorst [Bretthorst1990], successfully approximates even non-ideal peak shapes in the spectrum by constructing the model FID as a complex sum of as many exponentially decaying sinusoids as needed. A similar approach is taken by Indirect Hard Modelling but directly in the frequency domain [Kriesten2008]. These methods produce a convenient representation of a spectrum as a frequency-intensity table. However, if peaks of separate species overlap there is no clear physical basis for separating the contributions from each species to a given peak. This raises a challenging problem of assigning the fitted peaks to compute the concentrations of the chemical species, which often is the main goal of the analysis." Finally, we have added the following disclaimer to the Conclusions: "This paper considers model fitting approaches

only in the frequency-domain; it is not clear whether similar improvements would be obtained for time-domain methods."

Comment 2

In time-domain method-optimization using residual spectrum (FT of the residual FID) as can be misleading – particularly when dispersive residuals are used as a guide to the efficiency of amplitude estimation. Majority of the cases (in time-domain analysis) the dispersive residuals are the result of error in frequency estimation. For example, for a resonance of 1 Hz linewidth an error in frequency estimation by 0.02

Response

We agree that the dispersive residuals may often arise as a result of imperfect frequency estimation, however in our experience there are several other possible sources of model misspecification. As noted by the reviewer, it is likely that the dispersive contribution arising from frequency misspecification introduces a negligible error in the intensity estimate. However, that may not be the case for other sources of model misspecification, e.g., long range coupling effects which we do not normally consider. We found that phase and baseline adjustment in these cases allows us to improve the intensity estimates.

**Comments from the Second Reviewer**

The Second Reviewer raised the questions about the setting of the number of components and the definition of noise.
Comment 1

There is an example involving the modeling of a mixture in the manuscript which leads me to think that deconvolving mixtures is a supported feature. If this is correct, then perhaps the authors implicitly assume that the number of components are a known quantity? If so, then this assumption should be stated explicitly since the number of components in a linear mixture model in the presence of noise can have influence the calculated parameters of the model in a significant way.

On the other hand, if the authors assume that the number of components is an unknown parameter, then two additional points for discussion should be included in order to help readers develop a fuller picture: a) what procedures should be used for deriving the number of components?, and b) what would be the impact of the number components on the "model adjustment" process proposed by the authors?

Response

In our method, we assume that the chemical species in the mixture along with their ideal model signatures are known and available. This is a common assumption in many industrial applications dealing with routine analysis of similar mixtures (e.g. quality control and reaction monitoring). We have added the following sentences to clarify this point: "Instead, in our method we assume that the chemical species present in the mixture are known. This is often the case in many industrial applications concerned with routine analysis of similar samples, e.g. for quality control or reaction monitoring [Dalitz2012, Mitchell2014, Kern2018]. The ideal signature spectra of the analysed species are available, and we aim to adjust them to faithfully reflect the analysed data." in the Introduction and "Here we assume that the analysed components are known and K is fixed; model fitting with adjustable number of components has been previously explored in [Rubtsov2007]. If the experimental data contains any unexplained

components (observed as unfitted peaks in the residual spectrum), the model matrix needs to be extended to include these peaks before applying the proposed adjustment method." in Sec. 2.1.

Comment 2

It is worth noting that NMR signals are non-stationary, and there is supporting data for the view that the appropriate noise model for NMR spectra is not Gaussian. As a result, the outcome of the model fitting stage, depending on the objective function used for optimization, may lead to a residual noise background that is challenging to characterize. More specifically, the underlying distribution of this residual noise may change (according to the standard metric) in every iteration of an algorithmic optimization and at every point of the spectrum. This phenomenon, combined with lack of a priori knowledge about the number of components in the model, can hinder the adjustment procedure for weaker peaks.

Response

We agree with the reviewer that noise in NMR spectra is not necessarily Gaussian, though in many cases the approximation of Gaussian noise is remarkably accurate. Nevertheless, the reviewer is correct that if the noise is not Gaussian, it may make quantification of weaker peaks more challenging. As we note in the response above, we assume the number of components in the model is known a priori and this significantly reduces the challenge arising from non-Gaussian noise. Furthermore, our method is intended primarily for qNMR applications, where SNR in the spectra would normally be > 100. For these cases, the error introduced by assuming Gaussian noise appears to be minimal.

Comment 3

As stated in the manuscript, the model adjustment process attempts to ": : : find a model spectrum, which after the subtraction from the experimental data would lead to exclusively noise in the residual." This is an intuitively appealing objective, but only a heuristic (not rigorous) form of this statement is given. The authors state in a later paragraph that the specification of this objective through equation 5 is ill-posed, but their statement does not explicitly address the issue of how "exclusively noise" can be determined. Recall that two noise vectors of identical power (or different power) can be far apart in the standard metric but rather close in an appropriately selected metric. For example, the cosine distance for two noise vectors generated from the same random process can be close to 1 ($1 - r$, where $r$ is the cosine between two noise vectors, is considered very far). Therefore, the precise specification of how the residual becomes exclusively noise is necessary.

Response

As noted by the Reviewer, our goal for achieving "exclusively noise in the residual" is appealing but heuristic in nature, and this is exactly how we intended it to be perceived to motivate our approach. The objective of our method is to ensure the intensity of the peaks is obtained more accurately than if model fitting alone is used. It is not critical that the denoising works perfectly. In fact, the denoising step in our method is non-essential and can be omitted entirely; nevertheless, we found it particularly useful here. Furthermore, due to the dispersive nature of the residual (with high positive and negative peaks), it is relatively easy to denoise with the standard wavelet thresholding algorithm. However, as we noted in the paper, the choice of the denoising algorithm, as well as its parameters, is by no means exclusive and serves only as a guide for the reader. Other denoising methods can be applied as well, if desired (or this step can be

omitted altogether).

The following has been added to the paper to clarify these points: "... after subtraction from the experimental data would lead to exclusively noise in the residual. We achieve this heuristic goal by explicitly applying a denoising algorithm to the residual spectrum and then redistribute the remainder among the model signatures." and "We emphasize however that a multitude of denoising approaches exist and other methods (as well as different wavelet parameters) can be more suitable for a specific dataset. Furthermore, the denoising step in our method is not strictly necessary since the contribution of zero-mean random distortions asymptotically cancels out when the area under the residual is computed (as in the usual peak integration). However, we found it useful to include here to reduce the resulting uncertainty, especially when the number of points in the spectrum is not sufficiently high." in Sec. 2.3.

Comment 4

The Reviewer states: "The authors present a figure showing the impact of symlet order on RMSE in peak ratios which seem to indicate the superiority of the symlet over reference deconvolution. However, the RMSE in peak ratios seem to suggest very small improvements as a result of applying the proposed algorithm. Is this correct?".

Response

This is not correct. Fig. 11 compares the results of using parametric lineshape models (e.g. Gaussian, which is referred to as 2nd order symmetric shape) with the proposed adjustment approach. When the parametric lineshape models are used, the norm of the residual is reduced but this does not produce a significant improvement in RMSE of the mole fractions. The proposed adjustment procedure demonstrates clear benefits in the improved accuracy of the results, as has been noted in Sec. 4.1 and the caption

to the figure. The axis titles in Fig. 11 have been updated to avoid confusion.

Comment 5

Clear and concise statements regarding the assumptions made in devising this algorithm. For example, is it assumed that the number of components is known a priori?

Response

The reviewer is correct, we assume the number of components is known a priori. This is now stated in the manuscript, as noted under the response to point 1.

Comment 6

The capabilities and limitations of the approach. For example, how many peaks can be handled and how much overlap is tolerated by the algorithm?

Response

The capability and limitation of the approach is largely determined by the capability of the underlying model fitting algorithm. We have not yet tested our model with large numbers of components, but various related model fitting approaches have been applied to relatively large numbers of components, and so we anticipate our method should scale similarly well. The extent of overlap has been addressed in prior work for model signals, we are in the process of investigating this effect in real experimental systems. To clarify these issues, we have added the following to the manuscript: "Even though the above examples contain relatively low number of components they are representative of systems commonly encountered in industrial settings [Dalitz2012,

Mitchell2014, Kern2018]. Other works, notably [Krishnamurthy2013, Anjum2018], have demonstrated the possibility of applying modelling approaches to large numbers of components, and thus we expect our method to scale well. Furthermore, it has been shown that significant peak overlap can be tolerated in ideal artificial examples [Matviychuk2017], and thorough investigation of these effects in real-world systems is the topic of ongoing research."

Comment 7

Modes of parameters specification in the software, and guidance on the method of comparing outcomes will add. If it is being used as a user-guided tool, what utilities are available for guiding the user in selecting the "best" solution? If it is left to the user, then please state it. The use of software as a user-supervised tool. Do the authors view this as an automated tool, or is it viewed as a user-operated tool to gain insight? Availability of the software and license rights.

Response

The parameters used here have proven suitable for all applications where we have tested the method, but they are by no means exclusive. To clarify this point, we have added a comment at the end of Sec. 2.3 as follows: "We find the above settings to be suitable for all data sets we have tested, though in practice it is highly likely that for some samples the user would need to tune these parameters themselves (i.e. settings of the denoising algorithm and the width of the median filter). As such, in its default set up the algorithm is capable of automatic processing of typical high-field and benchtop spectra, however there exists a possibility for a more interactive processing approach if needed." Finally, we note that we are currently developing a Python package (including a GUI) for processing of NMR data in a partnership with several commercial users

of qNMR. We would welcome contributions from other parties, and we are open to collaboration with other potentially interested users.

---

## Author Response (AR2)

The Editor has made two comments.

1) The authors should state in the abstract that the method explcitly assumes that the number of components in the sample is known.

Response. The following sentence has been added in the Abstract:

5    "Specifically, we assume that the number of mixture components along with their ideal spectral responses are known..."

2) In the concluding remarks, the authors should address the fact that as a heuristic/ ad hoc approach, there is no formal guarantee that the accuracy reported by the authors will be recapitulated in a different signal/noise/complexity regime, and that use should be accompanied by empirical validation.

Response. The following sentence has been added in the Conclusions section:

[revised manuscript text omitted]